# Molecular basis for the production of cyclic peptides by plant asparaginyl endopeptidases

M.A. Jackson [1], E.K. Gilding [1], T. Shafee [2], K.S. Harris[2], Q. Kaas [1], S. Poon[2], K. Yap[1], H. Jia[1], R. Guarino[2], L.Y. Chan[1], T. Durek [1], M.A. Anderson [2] & D.J. Craik [1]

Asparaginyl endopeptidases (AEPs) are proteases that have crucial roles in plant defense and seed storage protein maturation. Select plant AEPs, however, do not function as proteases but as transpeptidases (ligases) catalyzing the intra-molecular ligation of peptide termini, which leads to peptide cyclization. These ligase-type AEPs have potential biotechnological applications ranging from in vitro peptide engineering to plant molecular farming, but the structural features enabling these enzymes to catalyze peptide ligation/cyclization rather than proteolysis are currently unknown. Here, we compare the sequences, structures, and functions of diverse plant AEPs by combining molecular modeling, sequence space analysis, and functional testing in planta. We find that changes within the substrate-binding pocket and an adjacent loop, here named the "marker of ligase activity", together play a key role for AEP ligase efficiency. Identification of these structural determinants may facilitate the discovery of more ligase-type AEPs and the engineering of AEPs with tailored catalytic properties.

[1] Institute for Molecular Bioscience, The University of Queensland, Brisbane, Queensland 4072, Australia. [2] Department of Biochemistry and Genetics, La Trobe Institute for Molecular Science, La Trobe University, Melbourne, Victoria 3086, Australia. Correspondence and requests for materials should be addressed to M.A.A.(M.Anderson@latrobe.edu.au) or to D.J.C.(d.craik@imb.uq.edu.au)

Plant asparaginyl endopeptidases (AEPs) typically act as master proteases controlling a myriad of cellular functions, most notably in plant defense responses to elicit programmed cell death[1], and as key processing enzymes of seed storage proteins[2]. In a small but growing number of angiosperms, AEPs play a very different role, namely in the post-translational cyclization of ribosomally synthesized peptides, including the 14 aa sunflower trypsin inhibitors (SFTI-1)[3] and the ~30 aa cyclotides[4]. Backbone cyclization of these peptides in planta is thought to enhance their stability, allowing high-level accumulation in tissues to deter herbivore feeding[5,6]. More generally, the cyclization of peptides for therapeutic applications is beneficial, resulting in improved stability and, in many cases, higher-potency over their linear counterparts[7]. Thus, the development of ligase-type AEPs as biotechnological tools to cyclize or ligate peptides or proteins would provide a valuable advance.

Unlike other plant defense peptides such as the defensins[8,9], cyclotides have a narrow distribution, found only in certain species of the Rubiaceae, Violaceae, Fabaceae, Solanaceae, and Cucurbitaceae[10–14]. Although cyclotide precursors differ between the five cyclotide-producing plant families, one common feature is a conserved asparagine or aspartic acid residue at the C-terminus of the cyclotide domain within the precursor (Supplementary Fig. 1a, b). This observation led to the hypothesis that AEPs, which by definition recognize asparagine or aspartic acid residues in the P1 position, might operate in reverse of their normal proteolytic function to biosynthetically cyclize cyclotides[15]. However, early studies using the non-native cyclotide-producing model tobacco plant Nicotiana benthamiana Domin as a host for cyclotide transgene expression resulted in predominantly mis-processed linear peptides, indicating that the cyclization function is not efficient for endogenous tobacco AEPs, and thus probably not conserved in all AEPs[15,16]. The inability of tobacco to efficiently make cyclic peptides was recently overcome via the co-expression of cyclization-capable AEPs sourced from the cyclotide-producing species Oldenlandia affinis (R&S) DC. and Clitoria ternatea L.[17]. The approach of co-expressing optimized processing enzyme with a desired substrate, dramatically improved in planta cyclic peptide yields and opens up opportunities to develop plants into efficient and versatile biofactories for the production of therapeutic cyclic peptides.

Despite the recent in vitro characterization of two evolutionary distant AEP ligases[18,19], little is known of the structural changes and possible convergent features present within ligase-preferring AEPs from different plant species. Mechanistically, it is likely that cyclization proceeds via a transpeptidation reaction and not through hydrolysis followed by peptide ligation. This has been elegantly shown in vitro using the recombinant O. affinis AEP ligase (OaAEP1$_b$)[18] and in situ using sunflower seed protein extracts[20], where a transpeptidation model is favored, with access of $^{18}$O-labeled $H_2O$ to the enzyme active site restricted. Unlike other peptide ligases, such as the cyanobactin macrocyclase PatG[21], where a large helix turn helix domain shields the active site from water, no obvious domains or insertions are identifiable in OaAEP1$_b$. Recently the crystal structure of the OaAEP1 zymogen (1 aa different to OaAEP1$_b$) was determined[22]. By comparison with the structure of a mammalian, protease-type AEP[23] it was revealed that OaAEP1 uniquely presents a shallow cleft that may allow N-terminal substrate residues access to the active site. However, it is not clear if this molecular feature is retained in the active enzyme and if it is characteristic of AEP ligases, or instead reflects differences between mammalian and plant AEPs. However, a single residue (Cys247) situated deep in the substrate pocket was found to be highly influential in ligase kinetics and thus was termed the "Gatekeeper" residue. Mutagenesis revealed that amino acid residues with small side chains at

this position are favorable for ligation efficiency, but residues involved in skewing AEP activity towards transpeptidation or hydrolysis were not identified. More recently, the structure of a sunflower AEP (HaAEP1) crystallized in its active form was determined[24]. In vitro reactions demonstrated a degree of ligase activity on an SFTI-1 substrate that could be improved marginally by mutating an asparagine residue (Asn73, HaAEP1 numbering), residing in close proximity to the catalytic Cys and His. Despite this improvement, the production of linear SFTI-1 predominated, suggesting that another unidentified sunflower AEP may play a more influential role, or that other factors are at play for cyclization of SFTI-1 in planta. Thus, an understanding of structural features that underpin the activity preference of AEPs is lacking.

Here we describe a comparative study of native and engineered plant AEPs, which combines functional in planta testing of AEP ligase activity[17], molecular modeling, and in silico protein space analysis. Despite family-specific cyclotide precursors and ligase competent AEPs occurring in families separated by many millions of years of evolution[25] (Supplementary Fig. 1a), several shared structural features are identified in ligase-type AEPs, explaining how these proteases have evolved independently to function as ligases. Our findings provide insights for both the discovery of AEP ligases and for rational engineering to improve AEP function for in planta and in vitro biotechnological applications.

## Results

**An AEP ligase from petunia.** To identify AEP ligases we first examined garden petunia (Petunia x hybrida E.Vilm.), the only plant of the Solanaceae family reported to naturally produce cyclotides (termed petunitides)[13]. We expressed the Oak1 precursor of kalata B1 (kB1, the archetypal cyclotide from O. affinis[26]) in petunia and discovered that petunia produced cyclic kB1, identical to native kB1 from O. affinis (Supplementary Fig. 2a, b). This contrasted with Oak1 expression in the cyclotide non-producer N. benthamiana, which produced predominantly mis-processed (linear) kB1 (Fig. 1a, b, Supplementary Fig. 2a)[15–17]. This led to the hypothesis that petunia has at least one ligase-type AEP that we set out to identify. RNA-seq data from P. x hybrida "Mitchell" shoots and leaves (Supplementary Table 1) facilitated the identification and cloning of four unique AEP gene sequences (Supplementary Fig. 3) that were named as PxAEP1, 2, 3a, and 3b, with the latter two isoforms displaying high amino acid similarity (91.3 and 90.5% pairwise identity across the propeptide and mature enzyme respectively). We used transient expression in N. benthamiana leaves to assess the ability of each petunia AEP to function as a ligase in planta. In the initial experiments we provided a ligation substrate by co-expressing the gene PET1, which encodes the native precursor of the petunia cyclotide phyb A[13]. However, no cyclic or linear Phyb A peptides were detected, even when the cyclotide precursor was co-expressed with the petunia AEPs. This might reflect an inability of this peptide to fold correctly in the endoplasmic reticulum of N. benthamiana leaf cells, which would result in rapid clearance and an inability to detect any Phyb A related peptide. As an alternative, we tested each of the four petunia AEPs with the kB1 precursor gene Oak1. Relative in planta cyclization efficiencies were determined by comparing MS signal intensities for cyclic kB1 compared to linear kB1 peptides upon co-expression of transgenes encoding the AEPs and the Oak1 substrate. The PxAEP3b isoform produced the most cyclic kB1 that represented an average 29.6% (±1.3% s.e.m. n = 12) of the total kB1-related MS signals compared to only 8.0% (±0.5% s.e.m. n = 14) cyclic kB1 for the no AEP control (Fig. 1b, c). Interestingly, the most closely related AEP, PxAEP3a, exhibited relatively little ligase

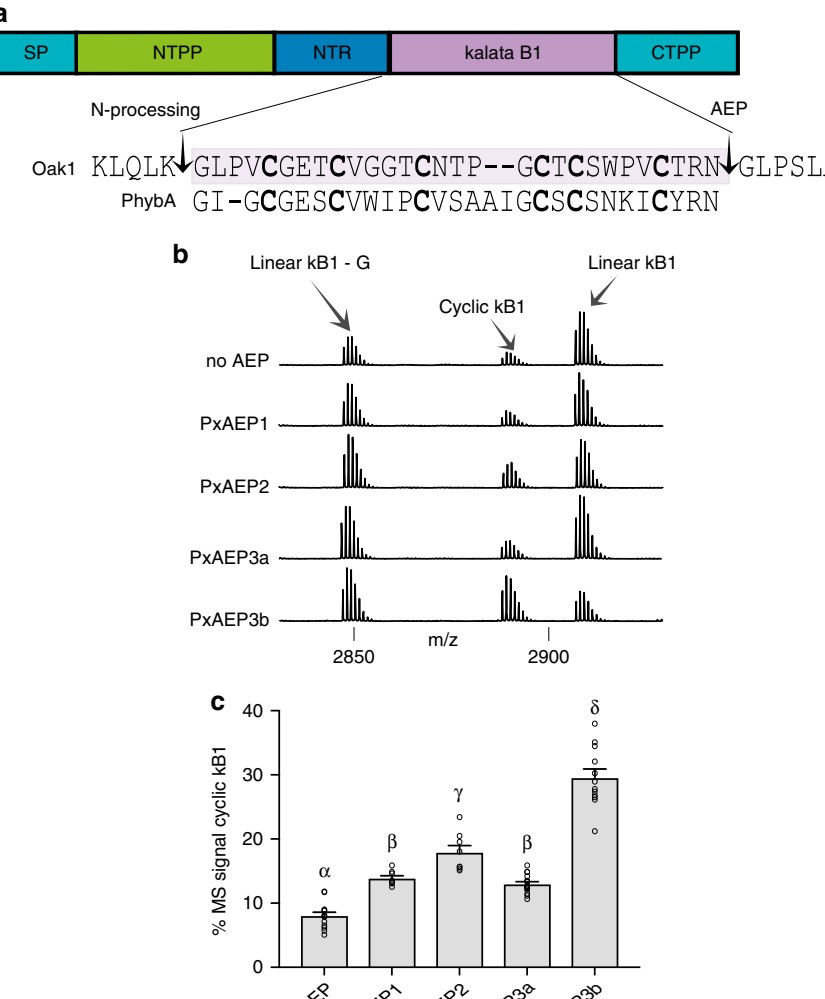

**Fig. 1** Petunia AEP isoform PxAEP3b functions as a ligase in planta. **a** Gene structure of *Oak1*, which encodes the prototypic cyclotide kB1 of *O. affinis*[26]. For maturation, the kB1 precursor is directed initially to the endoplasmic reticulum due to the presence of a signal peptide (SP). Here the precursor protein folds and disulfide bridges are formed. Further trafficking towards the vacuole facilitates resident proteases to release the amino terminal propeptide (NTPP) and amino terminal repeat (NTR). The liberation of the kB1 N-terminus is a pre-requisite for AEP-mediated backbone cyclization[52]. Shaded in purple is the 29 aa, six-cysteine containing kB1 peptide sequence. Flanking residues within the NTR and carboxyl terminal propeptide (CTPP) important for post-translational processing are shown. Aligned to kB1 is the Phyb A cyclotide from petunia, which could not be detected in *N. benthamiana* leaves despite co-expression of petunia AEP isoform PxAEP3b. Although similar in sequence, kB1 and Phyb A differ in topology with the presence of a cis-proline in kB1 leading to a conceptual twist in the cyclic backbone. **b** Representative MALDI-MS of kB1-related peptides produced upon transient gene expression in *N. benthamiana* leaves of *Oak1* with or without co-expression of petunia AEP genes. **c** Mean and s.e.m of the percent MS signals obtained for cyclic kB1 over the total MS signal of all kB1-related peptides. Treatments carrying unique Greek lettering are significantly different ($P < 0.05$) as determined by Tukey's ANOVA. Replicates are as follows: $n = 14$ (No AEP), $n = 8$ (PxAEP1 and PxAEP2), $n = 16$ (PxAEP3a) and $n = 12$ (PxAEP3b). Error bars are s.e.m.

activity (13.0% (±0.3% s.e.m. $n = 16$)) (Fig. 1b, c). This suggests that the small number of sequence polymorphisms (29/304 in the mature enzyme domain) (Fig. 2a, Supplementary Fig. 3) between PxAEP3a and PxAEP3b might be important for AEP functional preference.

The positions of the polymorphic residues between PxAEP3a (protease type) and PxAEP3b (ligase-type) were mapped using molecular modeling by threading the petunia AEPs onto the OaAEP1 zymogen structure (PDB accession 5HOI). The polymorphic residues were distributed throughout the enzyme (Fig. 2a−c), but of foremost interest in the ligase-type AEP were two amino acids near the substrate-binding pocket (polymorphisms 18 and 19) and two residues within a proline-rich loop (poly-proline loop) (polymorphisms 20 and 21)[27]. In addition, comparison of the ligase and protease revealed a 5 aa deletion in the ligase (polymorphisms 26–30). We tested the contribution of

these polymorphisms towards the ligase activity of PxAEP3b by preparing genes encoding chimeric PxAEP3s and examining their activity in our rapid in planta AEP assay with the kB1 precursor (Oak1) as substrate. To achieve this, varying numbers of PxAEP3b polymorphisms were swapped into PxAEP3a (Fig. 2c).

Initially we focused on the enzyme's substrate pocket, where in OaAEP1 the nature of the Gatekeeper residue (Cys247) has been reported to strongly influence ligase kinetics[22]. PxAEP3b (assigned as a ligase-type AEP) has a valine residue at this position (polymorphism 19) whereas PxAEP3a (a protease-type AEP) has a glycine (Fig. 2a, b, Supplementary Fig. 3). Additionally, polymorphism 18, which is three residues further away from 19, is a serine (PxAEP3a) to glycine (PxAEP3b) change. We hypothesized that the glycine/valine and the serine/ glycine polymorphisms in the ligase-type AEP PxAEP3b may underpin ability to cyclize kB1 in planta. However, when these

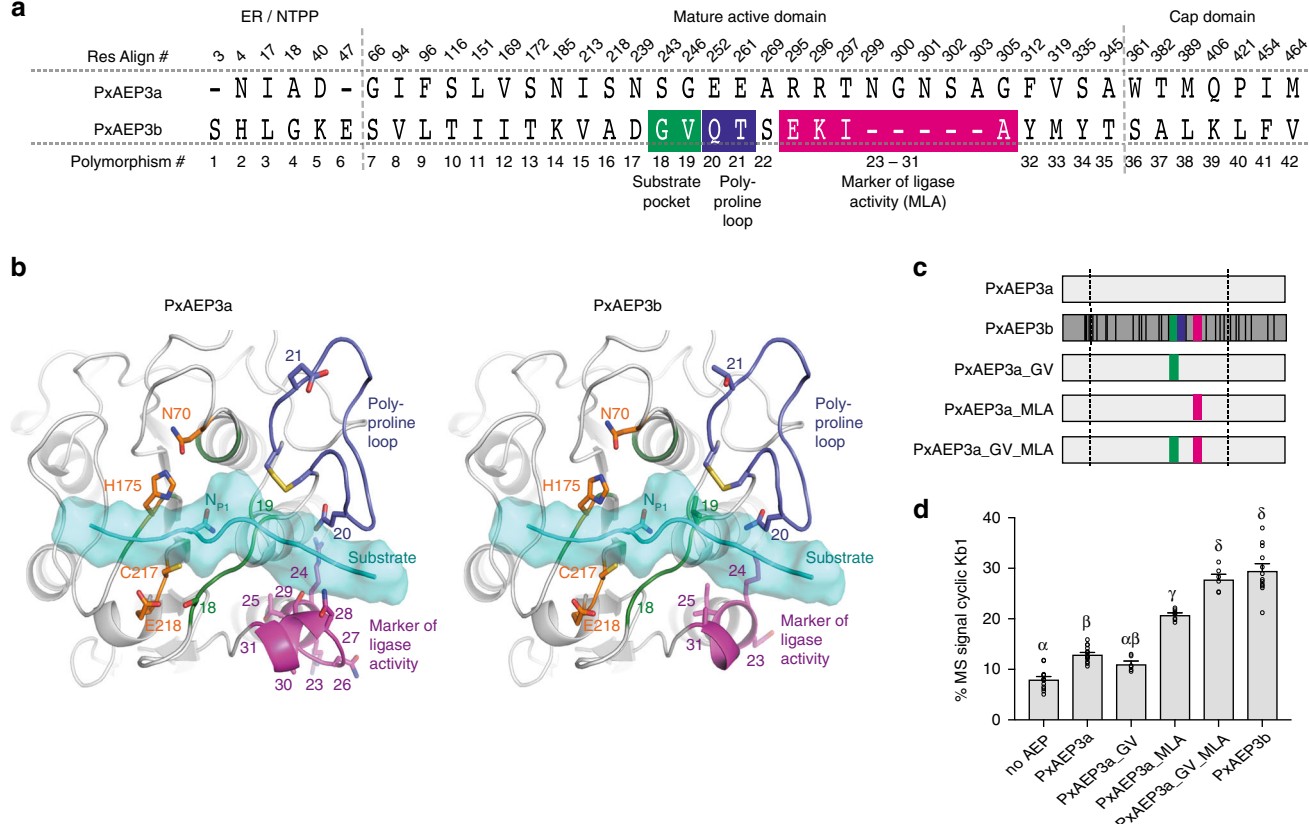

**Fig. 2** Identification and functional testing of the key structural features unique to the petunia ligase PxAEP3b. **a** Polymorphic residues between petunia isoforms PxAEP3a and PxAEP3b (see also Supplementary Fig. 3 for residue alignment). Those closest to the substrate pocket include polymorphisms 18 and 19 (green), polymorphisms 20 and 21 (blue) within the poly-proline loop and polymorphisms 23–31 (magenta) here named the marker of ligase activity (MLA). **b** Homology model of the substrate-binding site of the catalytic domain of PxAEP3a (left) and PxAEP3b (right). A substrate peptide bound in the binding site (cyan) was modeled by homology using the structure of human AEP (PDB ID: 4awa; www.rcsb.org) bound to a suicide substrate and extended at the C-terminus by a polyalanine peptide. The asparagine residue in the P1 pocket is noted $N_{P1}$. Polymorphic positions in and around the substrate pocket are shown as sticks, numbered and colored according to panel **a**. The catalytic residues are represented in orange sticks and numbered according to homologous residues on OaAEP1$_b$. **c** Schematics of petunia AEP genes prepared to test the influence of polymorphic residues on the in planta cyclization of the kB1 precursor. Colors used in the bars represent the same regions and colors in panel **a**. **d** MALDI- MS analysis of *N. benthamiana*-produced peptides upon co-expression of petunia AEP variants with *Oak1*. For all treatments, means and s.e.m. are of the percentage MS signal obtained for cyclic kB1 over the total MS signal of all kB1-related peptides produced. Treatments carrying unique Greek lettering are significantly different ($P < 0.05$) as determined by Tukey's ANOVA. Replicates are as follows: $n = 14$ (No AEP), $n = 16$ (PxAEP3a), $n = 6$ (PxAEP3a_GV), $n = 8$ (PxAEP3a_MLA), $n = 6$ (PxAEP3a_GV-MLA), and $n = 12$ (PxAEP3b). Error bars are s.e.m.

two residues were substituted into the PxAEP3a protease (construct PxAEP3a_ GV) and tested in planta there was no improvement in the ratio of cyclic to linear kB1 product (Fig. 2d).

We next determined whether the polymorphisms 23–31, which include the 5 aa deletion in the PxAEP3b ligase (Fig. 2a, Supplementary Fig. 3), contribute to ligase activity. For this we exchanged the polymorphic residues present in PxAEP3b into the PxAEP3a protease (PxAEP3a_MLA) and tested ligation activity in the in planta assay. This time there was a significant ($P < 0.05$, as determined by Tukey's ANOVA) increase in the cyclic to linear kB1 ratio relative to the PxAEP3a protease (Fig. 2d). Thus, this region contributes to the ability of PxAEPs to cyclize kB1 in planta leading us to name these polymorphic positions the "marker of ligase activity" (MLA). Nevertheless, the mean cyclic to linear ratio obtained was still lower than that observed when native PxAEP3b was expressed with *Oak1*, suggesting other independent or synergistic structural features contribute to ligase activity (Fig. 2b). We then prepared a construct, which combined the residue changes specific to the substrate pocket together with the MLA (construct PxAEP3a_GV_MLA). These changes improved the ratio of cyclic to linear products significantly

($P < 0.05$, as determined by Tukey's ANOVA) compared to experiments where these elements were tested alone (Fig. 2d). Indeed, the cyclic to linear ratio obtained was similar to that obtained upon expression of the native PxAEP3b, suggesting that the additional polymorphic residues that had not been altered (Fig. 2a) have only a minor role in the in planta cyclization of kB1 if any.

**The MLA is present in evolutionary distant AEP ligases.** Having established that the MLA and the substrate pocket residues of the petunia PxAEP3b isoform are important for the in planta cyclization of kB1, we examined whether these structural markers of ligase activity exist within evolutionarily distant AEPs that also function as peptide ligases. Alignments of AEP sequences were generated from diverse taxa in the cyclotide-producing plant families of the Solanaceae, Rubiaceae, and the Fabaceae with the human AEP (legumain) sequence included as reference (Fig. 3). The known AEP ligases from *O. affinis*, OaAEP1$_b$[18] and OaAEP3[17] had the same MLA deletion as PxAEP3b, but this deletion had not occurred in the protease-type

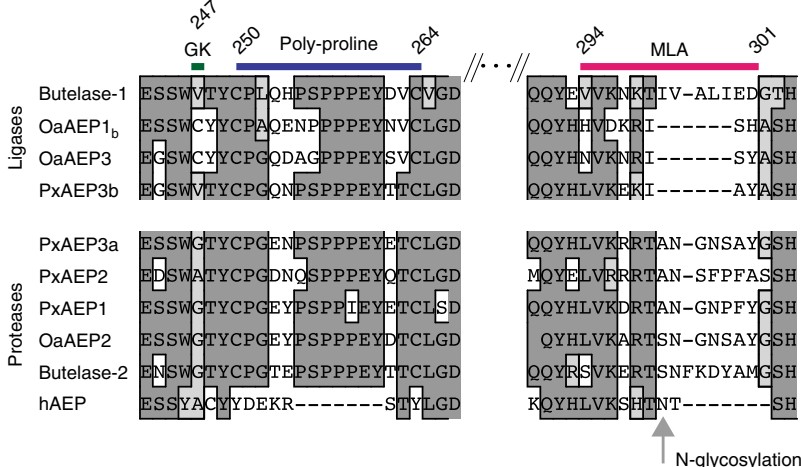

**Fig. 3** The marker of ligase activity (MLA) is present in evolutionary distant AEP ligases. Multiple sequence alignment of functionally verified AEP plant ligases from *O. affinis*, *C. ternatea* and *P.* x *hybrida* with protease-type AEPs from the same plant species. Overlaid by a magenta bar is the region defined as the marker of ligase activity (MLA), which contains a deletion in the functionally verified ligases from petunia (PxAEP3b) and *O. affinis* (OaAEP1b and OaAEP3) and is rich in hydrophobic residues in the case of the butelase-1 from *C. ternatea*. For human AEP (hAEP), a glycosylation site is present within a similarly small MLA region[28]. Overlaid by a green bar is the Gatekeeper (GK) residue (Cys247, OaAEP1b numbering) identified in OaAEP1 to be important for its in vitro ligase efficiency[22]. Overlaid in blue is the plant-specific poly-proline loop, which is present in both plant ligase and protease type AEPs but missing from the human AEP (hAEP)

AEP, OaAEP2, that failed to cyclize kB1 in the in planta assay[17]. This finding supports a relationship between the nature of the MLA and the ability of a given AEP to cyclize kB1 in planta. Interestingly, human legumain also carries a similarly sized deletion at the MLA, but the mature enzyme is known to be *N*-glycosylated at this site[28], which would substantially increase the size of this region. There are no glycosylation signals within the MLA of plant AEP ligases, and hence this region would probably remain small in size. The sole known ligase-type AEP from *C. ternatea* (butelase-1) does not have a deletion at the MLA, but this MLA is divergent in sequence and significantly more hydrophobic (Grand Average of Hydropathy (GRAVY) score + 0.555) than butelase-2 (score –1.075). Furthermore, the residue that occupies the Gatekeeper position orthologous to Cys247 in OaAEP1b is always divergent between ligase and protease-type AEPs from the same species (Fig. 3).

**The MLA influences the yield of cyclic peptides in planta**. To test the in planta functional significance of the MLA present in cyclizing AEPs across different plant families, we engineered butelase-1 and OaAEP1b to each contain the reciprocal MLA from the protease-type AEPs, OaAEP2 and butelase-2,[29] respectively (constructs OaAEP1b_MLA and butelase-1_MLA). For this, the MLAs were defined as the residues from His294 to His301 for the OaAEP chimeras (OaAEP1b numbering), and from Ile289 to Asp295 for the butelase chimeras (butelase-1 numbering) (Supplementary Figs. 4 and 5). The relative in planta cyclization efficiency of wild-type and MLA-modified AEPs was obtained by comparing MS signal intensities for cyclic kB1 compared to linear kB1 peptides upon co-expression of transgenes encoding the AEPs and the *Oak1* substrate (Fig. 4a). A modified *Oak1* gene was also tested (construct Oak1_SFTI, Supplementary Fig. 6) in which the kB1 domain had been replaced with the smaller 14 aa single disulfide containing SFTI-1 peptide. Control infiltrations of *Oak1* with native OaAEP1b and butelase-1 resulted in the expected high cyclic-to-linear peptide ratios reinforcing their role as ligases (Fig. 4a). In contrast, expression of both the MLA-reciprocated constructs resulted in significantly (*P* < 0.05, as determined by Tukey's ANOVA) reduced kB1 cyclic-to-linear ratios, consistent with the result obtained with the petunia AEPs. The importance

of the MLA for in planta kB1 production therefore extends to AEPs of different phylogenic origin. In the case of SFTI-1, OaAEP1b produced a cyclic product in planta (SFTI-1, 1513.7 m/z), with the relative yield significantly (*P* < 0.05, as determined by Tukey's ANOVA) decreased when the MLA-modified construct was used (Fig. 4b, Supplementary Fig. 6). Expression of butelase-1 or the MLA-reciprocated version did not result in any detectable cyclic SFTI-1, in agreement with the reported low efficiencies of plant-purified butelase-1 for activity at aspartic acid residues (P1) which is also reflected in the paucity of cyclotides from *C. ternatea* that contain Asp at the cyclization site[30,31].

To confirm the in planta data for the MLA-reciprocated enzyme (OaAEP1b_MLA), we produced recombinant OaAEP1b_MLA enzyme for in vitro analysis (Supplementary Fig. 7). We used mass spectroscopy (MS) to compare the activity of rOaAEP1b_MLA with wild-type rOaAEP1b by tracking the processing products of substrates (a modified cyclotide substrate kB1GI and an unrelated model substrate, R1AEP[18]) (Fig. 4c, Supplementary Fig. 8). Each enzyme produced predominantly cyclic product from each substrate, but the MLA mutant was much slower, and hence a substantial amount of the kB1GI precursor was still present after 72 h. This contrasted with rOaAEP1b, which completely converted the precursor substrate to cyclic product within 10 min. A similar observation was made with the R1AEP substrate, although the difference was not as pronounced (Supplementary Fig. 8). This suggests that the MLA does not directly impact the functional preference of the enzyme, but may facilitate more rapid processing of the kB1GI and R1AEP substrates, and thus directly influences the yield of cyclic peptide. In planta, this manifests as a decreased proportion of apparent cyclic product when a "protease-type" MLA is present (Fig. 4a, b).

**Protein sequence space modeling of ligase-preferred residues**. Having established that the presence of a ligase-type MLA has an important role in the yield of cyclic kB1 in planta, we next probed for residues that might influence the functional preference of an AEP more directly. We developed and implemented a protein sequence space analysis based on residue biophysical properties to compare 442 plant AEP sequences, including 6 known ligases, 12 known proteases, and 424 of unknown functions (Supplementary

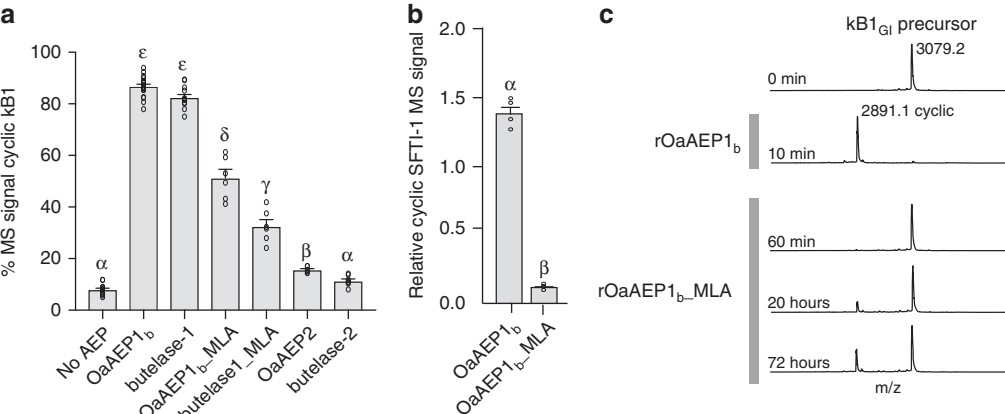

**Fig. 4** The MLA influences the yield of cyclic peptide in planta. **a** MALDI-MS analysis of *N. benthamiana*-produced peptides upon co-expression of AEP variants with *Oak1*. For all treatments, means and s.e.m. are of the percentage MS signal obtained for cyclic kB1 over the total MS signal of all kB1-related peptides produced. Treatments carrying unique Greek lettering are significantly different ($P < 0.05$) as determined by Tukey's ANOVA. Replicates are as follows: $n = 14$ (No AEP), $n = 16$ (OaAEP1$_b$), $n = 13$ (butelase1), $n = 9$ (OaAEP1$_b$_MLA), $n = 6$ (butelase1_MLA), $n = 6$ (OaAEP2), and $n = 8$ (butelase2). Error bars are s.e.m. **b** Relative abundance of cyclic SFTI-1 in leaf extracts after co-expression of *Oak1_SFTI-1* (Supplementary Fig. 6) with OaAEP1$_b$ ($n = 3$) or OaAEP1$_b$_MLA ($n = 3$). Error bars are s.e.m. Treatments carrying unique Greek lettering are significantly different ($P < 0.05$) as determined by Tukey's ANOVA. **c** In vitro comparison of the enzyme activity preference (protease vs ligase) of recombinant OaAEP1$_b$ and OaAEP1$_b$_MLA. Representative MALDI-MS profiles ($n = 3$) of the kB1 precursor (kB1$_{Gl}$) incubated with recombinant OaAEP1$_b$ and OaAEP1$_b$_MLA (each at 23.5 µg mL$^{-1}$ total protein) for up to 72 h. For rOaAEP1b, all precursor peptides were converted to cyclic kB1 (2891.1 m/z) within 10 min, while substantial precursor peptide remained in the case of OaAEP1$_b$_MLA even after 72 h of incubation. No linear kB1 peptide was detected from either enzyme. Observed monoisotopic masses (Da; [M + H]$^+$) are listed

Data 1, Supplementary Table 2). For this analysis, AEPs were considered as ligase-type if they had been tested in the in planta ligation assay with the *Oak1* substrate and produced cyclic-to-linear peptide ratio of >25%. AEPs were ranked as protease-type if the opposite was true, or if they had originated from plants previously shown to be inefficient at cyclizing the kB1 precursor. The known ligases included four from *O. affinis* (OaAEP1; OaAEP1$_b$: OaAEP3 and OaAEP4[17]), one from petunia (PxAEP3b; this study) and one from *C. ternatea* (butelase-1[17]) (Supplementary Table 2). Principal component analysis identified a set of covarying residue properties (Fig. 5a) that separated the known ligases from the known proteases with 96% accuracy (Fig. 5b, Supplementary Table 5). This led to the compilation of a set of 16 residues that best correlated with ligase activity (Fig. 5c, Supplementary Table 3). These 16 preferred residues included those closest to the active-site pocket, the Gatekeeper and its adjoining residue and residues within the poly-proline loop and the MLA (Fig. 5c, d).

We tested the validity of the modeling by introducing the ligase-predictive residues into the protease-type AEP, OaAEP2, to create OaAEP2(+) with the expectation of skewing the enzyme's functional preference from protease to ligase-type (Fig. 6a, b). As OaAEP2 already harbored two of the 16 ligase-preferred residues, only ten substitutions and four residue deletions were required. Both OaAEP2 and OaAEP2(+) were expressed heterologously in *Escherichia coli* and compared with OaAEP1$_b$ for ligase vs protease activity. The activity preference of each AEP was assessed in vitro using MS to track the processing products of a cyclotide precursor (kB1$_{GL}$) and the unrelated model peptide (R1$_{AEP}$) (Fig. 6c, d). The protease activity preference previously reported for rOaAEP2[17] was confirmed in the assay with R1$_{AEP}$, but it was inactive against the kB1$_{GL}$ substrate. In contrast, rOaAEP2(+) yielded a large proportion of cyclic product from each substrate, confirming its conversion to a preferential ligase. This demonstrates that substitution at key positions had skewed the activity preference of this native protease towards ligase activity, evidenced on two different substrates. However, native OaAEP1$_b$ remained the superior ligase producing the highest

proportions of cyclic product from each substrate. Although our modeling approach captured sufficient ligase-predictive residues to skew functional preferences on the tested substrates, there are clearly additional sequence elements and interactions responsible for full OaAEP1$_b$-like activity levels. Further expansion of characterized ligase/protease AEPs should enable refinement of the model to identify more features that enhance ligation activity.

Having improved the in vitro ligase activity of rOaAEP2 by substituting in ligase predictive residues (OaAEP2(+)), we next assessed whether the residues had individual contributions or acted in synergy with other residues (Fig. 6b). We used our plant-based cyclization assay to rapidly assess AEP performance of a series of OaAEP2 variants. We first designed a variant, named OaAEP2_select where, given the importance of the MLA, we extended the substitutions of residues into OaAEP2 to include OaAEP1$_b$ residues Asp295, Lys296, Ile 298, His300, and Ala301 (OaAEP1$_b$ numbering) (Fig. 6b, Supplementary Fig. 4). Additionally, we substituted Pro255 (OaAEP1$_b$ numbering) with an Asn instead of the Ala selected by the model. This kept all residue exchanges specific to those present in OaAEP1$_b$ and OaAEP2. By making these few adjustments we observed a significant ($P < 0.05$, as determined by Tukey's ANOVA) improvement in the cyclic-to-linear ratio of kB1 produced in planta with OaAEP2_select producing an average of 65.9% ($\pm 1.8\%$ s.e.m. $n = 14$) cyclic kB1 compared to OaAEP2+ that produced 33.9% ($\pm 1.0\%$ s.e.m. $n = 10$) (Fig. 6e). This improved in planta function was not due to major differences in transgene expression levels (Supplementary Fig. 9); thus, this finding reinforces that preserving the MLA region in AEP ligases is important for efficient in planta cyclization of kB1. To gain further insight we proceeded to make stepwise reversions of OaAEP2_select residues back to those in OaAEP2. Initially, the exchanged residues (OaAEP1$_b$ positions Asp161, Asp192, Glu314 and Gly316) that are most distant from the active-site were reverted (making construct OaAEP2_pocket_poly_MLA). This resulted in a small decrease of in planta cyclization efficiency (Fig. 6e). This left the polymorphisms closest to the active-site, (Cys247, Tyr248), the poly-proline loop (Gln253, Asn255, and Val263) and those substituted around the

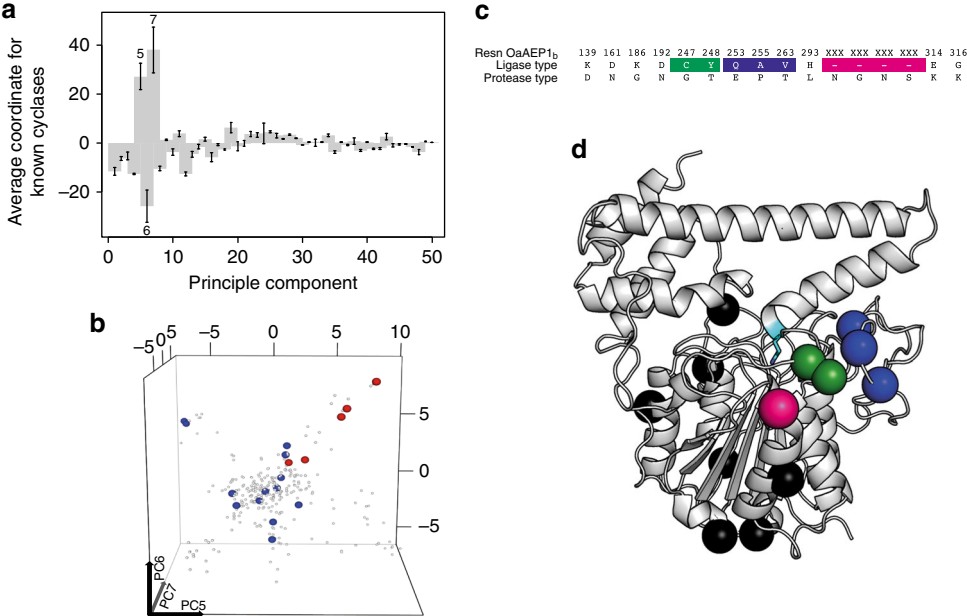

**Fig. 5** Plant AEP protein sequence space modeling. **a** Average protein sequence space coordinates of the six known ligases (Supplementary Data 1, Supplementary Table 2). **b** Plant AEP protein sequence space profiles after principle component analysis. Each point indicates a sequence, with its coordinates in the space determined by its residue properties. Axes are principle components 5, 6 and 7, which showed significant separation of known ligases (red) and proteases (blue) (Supplementary Table 2). **c** The 16 ligase predictive residues and positions on OaAEP1b. Color shading used is the same as in Fig. 2a. **d** Structure of the zymogen OaAEP1 (PDB accession 5HOI) with positions of ligase-preferred residues displayed. Residues shown in green include those closest to the substrate pocket, which includes the Gatekeeper residue Cys247 and the adjacent Tyr248. Those in blue are situated within the plant AEP-specific poly-proline loop, while the magenta colored sphere represents the position of the 4-amino acid deletion in the MLA. The seven spheres in black represent the residues most distal from the catalytic active site

MLA (His293, Asp295, Lys296, Ile298, His300, and Ala301) to consider. Reversion of the poly-proline loop polymorphisms (construct OaAEP2_pocket_MLA) had no impact on the cyclic-to-linear ratios. However, when the poly-proline loop polymorphisms were reverted in combination with the MLA or pocket residues, cyclic-to-linear ratios decreased significantly ($P < 0.05$, as determined by Tukey's ANOVA) (construct OaAEP2_pocket and construct OaAEP2_MLA respectively). Together these results indicate a degree of cooperativity between the MLA, the residues near the active site and the poly-proline loop.

**The MLA as a diagnostic marker of ligase-type AEPs.** The AEPs thus far assigned as ligase-type (as determined by the ability to cyclize kB1 in planta) have MLAs of two types: truncated or rich in hydrophobic residues. We therefore devised an approach using the MLA as a first pass marker for the identification of additional plant AEP ligases. The 40 aa stretch centered on the MLA of OaAEP1b was used to query NCBI and Phytozome databases using BLASTP, from which 281 AEP sequences were retrieved for analysis. We then employed two rules (in the presented order) to tag a sequence as a potential ligase: (1) the presence of a minimal MLA or an MLA harboring a hydrophobic patch (grand average of hydropathy (GRAVY) score > 0.0) and (2) the absence of any glycosylation signals within the MLA that would increase the loop size. Hits meeting these two criteria were then scrutinized for the presence or absence of the ligase-preferred residues identified in the protein space analysis. Using these rules, we identified 16 previously untested AEP sequences that passed rule 1, of which nine carried a potential glycosylation site (Supplementary Table 4). The remaining seven AEP sequences belonged to *Helianthus annuus* L. (sunflower), *Spinacia oleracea* L. (spinach), *Beta vulgaris* L. (beetroot), *Theobroma cacao* L. (cocoa tree), and

*Gossypium raimondii* Ulbr. (diploid Peruvian cotton). Of particular interest, due to the economic importance of cotton was the predicted ligase (named here GrAEP1; Gorai.009G046800.1; XP_012448355.1) from *G.raimondii*. An alignment with all other AEPs from *G. raimondii* revealed a unique minimal MLA in GrAEP1 (Supplementary Fig. 10). The ligase-predictive residues that were identified in the protein space modeling of the other plant AEP ligases were not conserved in GrAEP1. However, GrAEP1 has an alanine at the Gatekeeper site (A244) in contrast to a glycine in this position in the other four co-aligned *G. raimondii* AEP sequences. Subsequently, we tested the function of GrAEP1 in planta by transient co-expression with *Oak1*, and demonstrated a moderate level of ligase-type activity (27.5% ± 0.3% s.e.m. $n = 6$) (Fig. 7c). As no cyclotide-like sequences have been identified in the cotton genome, the biological relevance of an AEP ligase in cotton remains unknown.

Finally, we tested our ligase prediction rules as a means to rapidly identify ligase-type AEPs from known cyclotide-producing species. Unlike other cyclotide-producing plant families, all tested plant species within the Violaceae harbor cyclotides[14]. Despite this ubiquity no Violaceae AEP ligases have been identified. We used RNA-seq to obtain transcriptomes of the Violaceae member *Hybanthus enneaspermus* F.Muell from shoot and root tissue (Supplementary Table 1). Assembly of the data enabled the identification of three full-length AEP containing transcripts, which were subsequently cloned and named HeAEP1, 2, and 3 (Supplementary Fig. 11). Of the three, we predicted HeAEP3 to be a ligase due to the presence of a minimal MLA region (Fig. 7a, Supplementary Table 6). The ligase activity of HeAEP3 was validated by our in planta kB1 cyclization assay (Fig. 7b), where co-expression of HeAEP3 and *Oak1* resulted in an average of 77.9% (±1.4% s.e.m. $n = 8$) of the MS signals representing cyclic kB1. Thus, the presence of the minimal MLA in HeAEP3 was a clear predictor of AEP function.

## Discussion

Plant AEPs that have the unusual function of favoring peptide cyclization over hydrolysis are of high interest for their potential as biotechnological tools to stabilize peptides for therapeutic or agricultural applications[7]. However, the structural features that define ligase-type plant AEPs were until now largely unknown. Here, through bioinformatics, molecular modeling, and functional assays, we have identified orthologous positions across four cyclotide-producing plant families that have converged towards states that contribute to AEP ligase activity, and use this information to identify a further two AEP ligases and to confer ligase-type activity to protease-type AEPs. These ligase-predictive residues represent hot spots for further rational engineering to potentially broaden substrate specificity and improve catalytic efficiency, and markers to fast-track the discovery of additional AEP ligases.

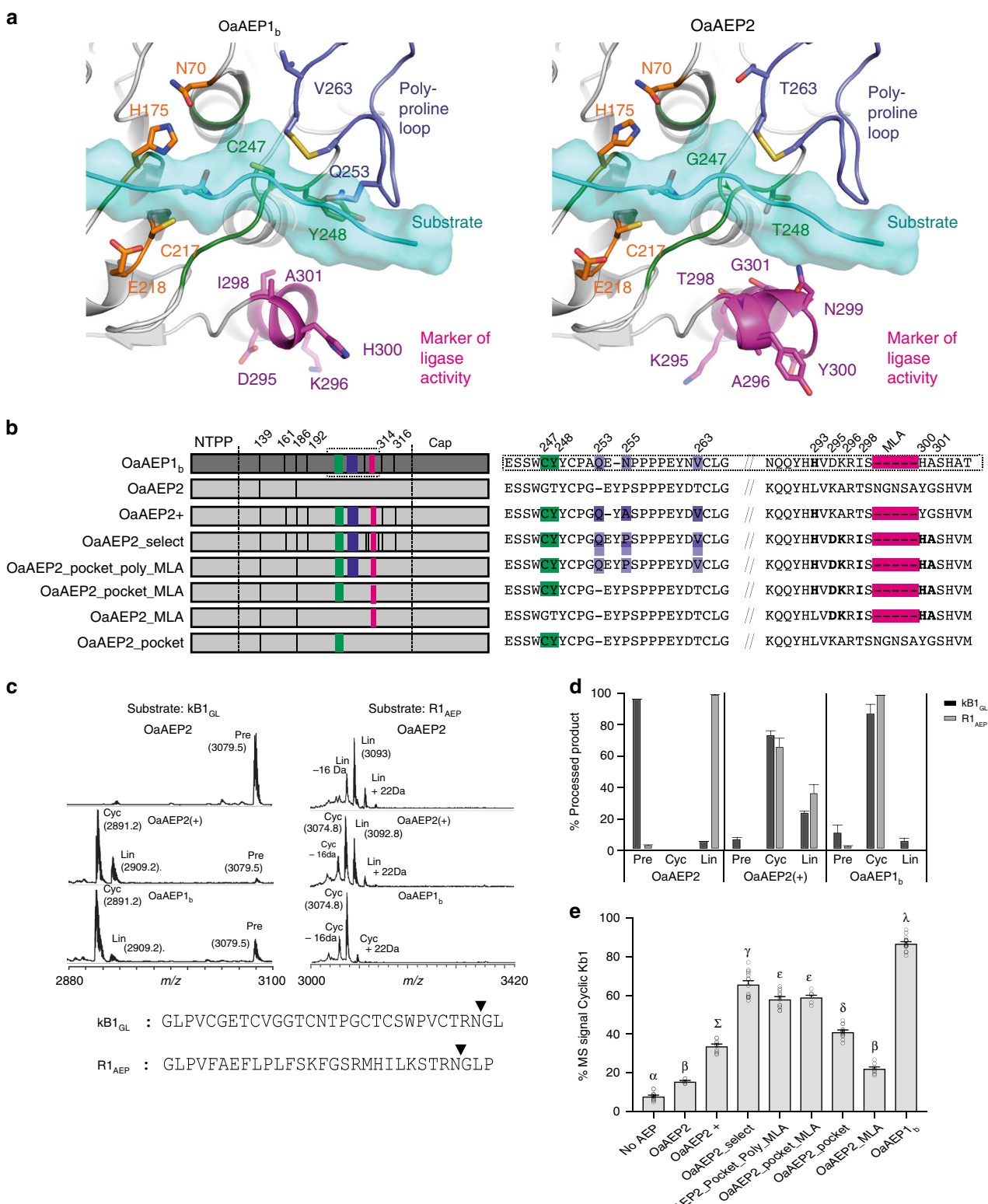

Unlike the capping helices necessary for the ligase activity of another type of peptide ligase, PatG[21], there are no readily identifiable structural or sequence markers that distinguish AEP ligases from AEP proteases. Our data have revealed that the switch in AEP activity from protease to ligase does not arise from a single mutation, deletion or insertion, but rather a combination of concerted changes. By leveraging the inherent sequence diversity of plant AEPs in combination with functional information, we identified a set of 16 covarying residues with properties that correlated with AEP functional preference (Fig. 5, Supplementary Table 3). Remarkably, the introduction of these 16 ligase-predictive residues into an otherwise protease-type AEP was sufficient to create a preferential ligase, as assessed on two unique precursors (Fig. 6c, d). Among these predictive residues is the previously reported Gatekeeper residue and its adjoining

residue[22], three residues within the plant-specific poly-proline loop, the MLA and seven distally located residues. Mutagenesis of limited subsets of these ligase-predictive residues revealed that changes at the MLA and residues at or adjacent to the Gatekeeper residue were most important for the in planta cyclization efficiency of kB1 (Fig. 6e). This is consistent with previous analysis of function-switching mutations[32,33], where the most influential residues are enriched around the active site. The significance of these residues is supported by the observation that they appear to have converged towards the same or similar amino acids at homologous positions in AEP ligases from evolutionarily distant plant families (Supplementary Fig. 12, Supplementary Table 6). Specifically, the MLA, either as a truncated- or a hydrophobic-type represents a clear departure from the common MLA type (extended and hydrophilic) and has occurred in ligase-type AEPs from distantly related families.

Although the exact role of the MLA in peptide ligation may only be realized upon further structural studies, we show here that it is essential for obtaining high yields of cyclic peptides in planta. By analogy to the structurally related mammalian caspases a role for the MLA in substrate selection is possible, with the MLA corresponding to the caspase c381 loop, a region known to influence substrate preference of mammalian caspases[34]. A decrease in substrate affinity of the MLA-reciprocated OaAEP1$_b$ (construct OaAEP1$_b$-MLA) would explain the overall decrease in cyclic peptide yields in planta as well as the apparent slow conversion of substrates in vitro. As no linear mis-processed peptide products were detected in the in vitro experiments, it is unlikely that the MLA itself contributes to the functional preference of the AEP. Interestingly in the case of the c381 loop of caspases, the loop is highly variable in size. Larger loops are present in caspase isoforms 3 and 7, where they narrow the $S_4$ pocket and alter the $P_4$ residue preferences of these isoforms[35]. Interestingly, the $P_4$ residue on incoming cyclotide substrates is, with the exception of those from M. cochinchinesis, invariably a cysteine residue (Supplementary Fig. 1b) and may play a role in the structural evolution of MLAs present in ligase-type AEPs. Despite this possibility, both recombinant OaAEP1$_b$ and plant-purified bute-lase-1 cyclize a variety of peptide and protein substrates in addition to cyclotide precursors, suggesting that the presence of a ligase-type MLA does not preclude cyclization of alternative substrates[19,36].

Recently, two additional plant AEP crystal structures were determined, sunflower HaAEP1[24] and Arabidopsis (AtLEGγ)[37]. These AEPs do not display a ligase-type MLA, suggesting that these AEPs would prefer proteolysis over ligation. Indeed, the recombinant enzymes were shown to be inefficient at in vitro macrocyclization or transpeptidation assays when conducted at

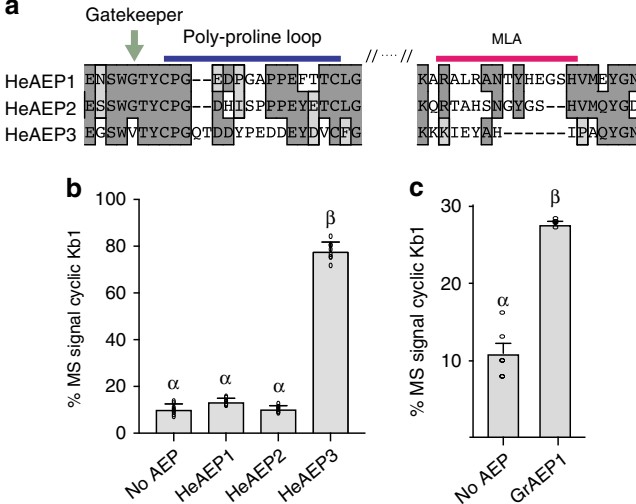

**Fig. 7** The MLA as a diagnostic marker for the discovery of ligase-type AEPs. **a** Sequence alignment of H. enneaspermus AEPs (see also Supplementary Fig. 11). HeAEP3 was predicted to be a ligase due to the truncation within the MLA and the presence of a unique Gatekeeper residue. **b, c** MALDI-MS analysis of N. benthamiana-produced peptides upon co-expression of (**b**) HeAEP1, 2 and 3 with Oak1 and (**c**) GrAEP1 with Oak1. For all treatments, means and s.e.m. are of the percentage MS signal obtained for cyclic kB1 over the total MS signal of all kB1-related peptides produced. Treatments carrying unique Greek lettering are significantly different (P < 0.05) as determined by Tukey's ANOVA. Replicates are as follows: n = 8 (no AEP, HeAEP1, HeAEP2, HeAEP3) and n = 6 (GrAEP1). Error bars are s.e.m.

**Fig. 6** Functional testing of ligase-preferred residues. **a** Homology model of the substrate-binding site of the catalytic domain of OaAEP1$_b$ vs OaAEP2. Substrate peptide bound in the binding sites (cyan) was modeled by homology using the structure of human AEP (PDB accession 4awa) bound to a suicide substrate and extended at the C-terminus by a polyalanine peptide. Residue positions within OaAEP2 that were replaced with corresponding residues of OaAEP1$_b$ to make the variants AEP OaAEP2+ and OaAEP2_select are indicated in panel **b** with those closest to the active site displayed in the models. The substrate-binding site (green), poly-proline loop (blue) and marker of ligase activity (MLA, magenta) are shown in sticks and numbered according to OaAEP1$_b$ numbering while the catalytic residues are represented in orange sticks. **b** Schematics of OaAEP variant genes prepared to test the influence of ligase-preferred residues on in planta cyclization efficiency. Color shading used is the same as in Fig. 2a. **c** Representative MALDI-MS spectra (n = 3) of the kB1$_{GL}$ and R1$_{AEP}$ precursor peptides following incubation with recombinant AEPs. rOaAEP2, 140 µg mL$^{-1}$ total protein, 22 h incubation; OaAEP2(+) 19.6 µg mL$^{-1}$ total protein, 2 h incubation; OaAEP1$_b$ 19.6 µg mL$^{-1}$ total protein, 5 h incubation. The filled triangles denote rOaAEP1$_b$ cleavage/ cyclization sites. Observed monoisotopic masses (Da; [M + H]$^+$) are listed. +22 Da and −16 Da peaks present in some precursor and product spectra are likely to represent Na$^+$ adducts and a synthesis-derived modification respectively. **d** The average (n = 3, s.e.m.) proportion of enzymatic processing products following incubation with recombinant AEPs. Cyc cyclic product, Pre linear precursor, Lin linear product. **e** MALDI-MS analysis of N. benthamiana-produced peptides upon co-expression of AEP variants with Oak1. For all treatments, means and s.e.m. are of the percentage MS signal obtained for cyclic kB1 over the total MS signal of all kB1-related peptides produced. Treatments carrying unique Greek lettering are significantly different (P < 0.05) as determined by Tukey's ANOVA. Replicates are as follows: n = 14 (no AEP), n = 6 (OaAEP2, OaAEP2_pocket_MAL and OaAEP2_pocket), n = 10 (OaAEP2 + ), n = 14 (OaAEP2_select), n = 12 (OaAEP2_pocket_poly_MLA), n = 8 (OaAEP2_MLA), and n = 22 (OaAEP1$_b$). Error bars are s.e.m.

pH 5.0, which corresponds to the plant vacuolar environment where plant ligase-type AEPs are present during normal physiological conditions[24,37]. Interestingly, when the pH was increased to 6.5, peptide ligation activity significantly improved. Nevertheless, an in planta role for a neutral pH ligase-competent AEP is yet to be identified, and it was speculated that this behavior might be important during programmed cell death[37] when vacuole resident AEPs could permeate into the more pH neutral cytoplasm[38]. Our present work focused on AEPs sourced from cyclotide-producing species, for which AEP-mediated cyclization of cyclotides is thought to occur in acidic vacuolar compartments[39,40]. We indeed observe a biological role for unique AEP isoforms that have co-evolved for efficient peptide ligation and cyclization. It remains unclear if cyclotide-producing plant species produce protease-type AEP that could display an increase in ligase-activity at neutral pH; however, the MLA determinant that we identified seems to be only relevant for predicting ligase-type activity in the vacuole environment of plants.

Beyond its functional role, the MLA represents an easily identifiable marker of AEPs capable of cyclizing kB1. Here by using the MLA as a diagnostic tool, we were able to rapidly detect an AEP ligase from the Violaceae plant family and have identified leads from a selection of non-cyclotide-producing plant families. Further discoveries will no doubt enhance the ligase toolkit available and offer rational engineering approaches for the ever-emerging applications of plant-derived AEP ligases.

## Methods

**Plant cultivation and material**. For producing material used in RNA-seq experiments, *P. x hybrida* cv "Mitchell" was cultivated in pots with UC mix under glasshouse conditions (Brisbane, QLD, Australia), or in the case of *H. enneaspermus* (original accession collected from a roadside near the Calliope river in Beecher, QLD, Australia) cultivated in UC mix in a growth room at 28 °C under 160 μmol of LED illumination (AP67 spectra, Valoya Oy, Helsinki, Finland). All in planta experiments with *N. benthamiana* and *P. x hybrida* "Mitchell" were cultivated in Jiffy pellets at the same temperature and under the same LED light banks as per *H. enneaspermus*.

**RNA-seq, transcriptome assembly, and analysis**. RNA was extracted from cultivated *H. enneaspermus* roots and shoots with expanded leaves, and *P. x hybrida* "Mitchell" leaves using TriZOL (Life Technologies, Carlsbad, CA, USA) according to the manufacturer's protocol. RNA samples passing quality control criteria for the Illumina RNA-seq pipeline at Australian Genome Research Facility (AGRF) were sequenced on a HiSeq 2500 system. Reads were analyzed with FastQC[41] before trimming of additional adaptor sequence from ends with trimmomatic[42] and assembly with Trinity[43] software as detailed in Gilding et al.[30], except with Trinity v2.0.6 with options for kmer size of 30 and normalization by read set.

**Vector construction**. DNA encoding either AEPs (with the exception of GrAEP1) or peptide precursors were recombined into the plant expression destination vector pEAQ-HT-DEST1[44] using Gateway® LR Clonase® technology. DNA encoding GrAEP1 was synthesized by Genscript and cloned into the plant expression vector pBIN19. Sequence verified vectors were transformed into *Agrobacterium tumefaciens* (strain LBA4404) by electroporation. Sequences used that have GenBank accession numbers assigned include: *Oak1* (AF393825), OaAEP1b (KR259377), OaAEP2 (KR259378), Butelase-1 (KF918345), and Butelase-2 (KR912009). AEP chimeric genes (Supplementary Figs. 4 and 5) and modifications of the *Oak1* gene (Supplementary Fig. 6, 13) were produced using a variety of molecular biology techniques including site-directed mutagenesis and Gibson assembly (Q5 Mutagenesis and HiFi NEBuilder kits respectively, NEB USA) and/or gene synthesis (IDT, Singapore).

**Transient expression in plant leaf**. Agrobacterium (LBA4404) cultures harboring AEP expression vectors and peptide precursor genes were grown in Luria-Bertani media to stationary phase before centrifugation and resuspension in infiltration buffer (10 mM MES (2-[N-morpholino]ethanesulfonic acid) pH 5.6, 10 mM MgCl2, 100 μM Acetosyringone). Each culture was mixed 1:1 at an OD600 of 0.5 before syringe leaf infiltration into *N. benthamiana* or *P. x hybrida* cv "Mitchell" plants at 5–6 weeks of age. At 6 days post infiltration, uniform-sized leaf discs were collected from a minimum of six replicate plant infiltrations and harvested for peptide analysis. For SFTI-1 expression, a third Agrobacterium culture was added

to the infiltration mix that contained an expression vector encoding a truncated kb6 peptide (Supplementary Figure 13). Expression of this latter construct resulted in the production of a kB6 peptide precursor without the required C-terminal residues for AEP-mediated cyclization. Thus, the accumulation level of linear kB6 was able to be used to normalize the resultant cyclic SFTI-1 MS signal obtained for each AEP tested.

**Peptide analysis and relative quantification**. Leaf discs (approx. 20 mg FW tissue) were placed in microfuge tubes with a ball bearing and ground to a fine powder in liquid nitrogen using a GenoGrinder. Peptides were extracted in 200 μL of aqueous (50% (v/v) acetonitrile, 1% (v/v) formic acid) with gentle mixing overnight. After centrifugation, the peptide containing supernatant was diluted 1:5 with 1% (v/v) formic acid before being desalted and concentrated using C18 Zip-Tips (Millipore). Samples were then mixed 1:1 with α-cyano-4-hydroxycinnamic acid (5 mg mL$^{-1}$ in 50% acetonitrile/0.1% TFA/5 mM (NH4)H2PO4) before being spotted and dried onto a MALDI sample plate for matrix-assisted laser desorption/ionization (MALDI)-time of flight (TOF) MS using an Applied Biosystems 4700 TOF-TOF Proteomics Analyzer. To enable relative quantification of cyclic and linear peptides, the sum of the isotope cluster area corresponding to each kB1-related peptide was taken as 100% of the expressed peptide.

**Transgene-derived kB1 purification and NMR spectroscopy**. Peptides were extracted from mature leaf tissue from an *oak1* expressing petunia cv "Mitchell" plant line using 50% (v/v) acetonitrile,1% (v/v) formic acid as solvent. Extracted peptides were purified by reverse-phase high performance liquid chromatography (RP-HPLC) using a Shimadzu LC-20AT HPLC system equipped with a Phenomenex C18 reverse-phase HPLC column (250 × 20 mm, 300 Å, 15 μm particle size). Dried extract was re-dissolved in 0.05% (v/v) trifluoroacetic acid/water (solvent A) followed by peptide isolation using a linear 1%/min gradient in 20–80% solvent B (0.45% trifluoroacetic acid/90% acetonitrile). Fractions were collected manually and monitored at two different wavelengths (214 and 280 nm) and masses were confirmed by MALDI- MS. Purity of peptides were examined using an ultra HPLC system (Shimadzu LC-30AD) and pure fractions were freeze-dried. For NMR analysis, freeze-dried peptide with high purity was dissolved in 90% H2O/10% D2O with the pH adjusted to 5.5. 4,4-dimethyl-4-silapentane-1-sulfonic acid was added as a chemical shift reference. All spectra were acquired on a Bruker Avance 600 MHz spectrometer at 298 K, with one-dimensional analysis followed by two-dimensional (TOCSY and NOESY) analysis. All NMR assignments were done using CCPNMR program version 2.4.0 [45].

**Molecular modeling**. Molecular models of petunia AEP isoforms PxAEP3a and PxAEP3b and *O. affinis* OaAEP2 were created by homology with the crystal structure of *O. affinis* OaAEP1 (PDB ID:5h0i) using Modeller9.18[46]. Side chain orientations of positions interacting with the substrate peptide were modified using PyMol according to those found in the crystal structure of human AEP bound to a suicide substrate (PDB ID 4awa[23]). A substrate peptide was generated by transferring the coordinates of the substrate from the 4awa crystal structure, and the peptide was then extended at the C-terminus by three alanine residues in Pymol. The model was then energy minimized using Gromacs 2016 [47].

**Sequence gathering, alignments and hydrophobicity scoring**. AEP sequences were gathered by Delta-BLASTp,[48] starting with the AEPs from *Petunia* x *hybrida* and *O. affinis* and limiting the search to only within the *Viridiplantae*. Retrieved sequences were trimmed to include only the mature C13 protease-like domain (MEROPS family classification) and those with >90% redundancy were removed, resulting in 442 sequences. Sequences were aligned by ClustalΩ using the BLOSUM62 substitution matrix and default gap penalties. For quantifying the hydrophobicity of the MLA region an online GRAVY prediction software was used and available at http://www.gravy-calculator.de/.

**Sequence space analysis**. The multiple sequence alignment (MSA) of 442 AEP sequences was numericised by describing each residue in the MSA by its R-group molecular weight (Daltons), net charge (Coulombs), hydrophobicity (Doolittle index)[49], disorder propensity (TOP-IDP)[50], and occupancy (binary descriptor). Values were normalized within each property and gaps were given the average column value for each property (other than occupancy). Each sequence was therefore described by a vector in a highly multidimensional space. The numericised protein sequence space was analyzed using Principal Components Analysis to summarize the main covarying sets of properties (using [R] prcomp)[51]. The resulting principal components describe covarying sets of residue properties. Three covarying residue property sets (principal components 5, 6, and 7) showed significant separation of known ligases and protease-type AEPs. The most highly weighted residue properties for those principal components were for 16 residues and the ideal ligase residue was identified for each position. A simple scoring metric based on the match to the ideal residue at the 16 positions, and ignoring all other positions, was used to rank the sequences. The known ligases had higher match scores than known protease-type AEPs, and a ≥25% match threshold correctly identified all the known ligases and excluded 94% of the known protease types.

**Expression and characterization of recombinant AEP**. Recombinant OaAEP1$_b$, OaAEP2, and OaAEP2+ without the putative signal peptide domains were produced as ubiquitin fusion proteins such that the final construct was His6-ubiquitin-OaAEP[18]. Bacterial lysate containing each AEP was passed over an anion exchange resin and eluted using a salt gradient. AEP-positive fractions were identified by western blotting using rabbit serum raised against OaAEP1$_b$. The AEP zymogen was self-activated by incubation at pH 4.5 (5 h; 37 °C) in the presence of reducing and chelating agents. The active enzyme was then captured by cation exchange and eluted using a salt gradient. AEP-positive fractions, as determined by activity against an internally quenched fluorescent peptide substrate, were pooled. The total concentration of protein was estimated by BCA assay according to the manufacturer's instructions. For the comparison of rOaAEP1$_b$ with rOaAEP1$_b$_MLA, recombinant AEP was produced in a similar fashion but without the ubiquitin domain fusion (Supplementary Fig. 7).

Linear target peptides (50–280 μM) were incubated with recombinant AEPs (19.7–140 μg mL$^{-1}$ total protein as indicated in the figure legends) in activity buffer (50 mM sodium acetate, 50 mM NaCl, 1 mM EDTA, 0.5 mM TCEP, pH 5) for up to 72 h at room temperature. The reaction mixture (5–10 μL) was desalted using C18 zip tips and eluted in 4 μL 75% acetonitrile, 0.1% trifluoroacetic acid (TFA). Eluted samples were mixed 1:4 with the MALDI matrix α–cyano-4-hyroxycinnamic acid, spotted onto a MALDI plate and analyzed by an Ultraflex III TOF/TOF (Bruker) in positive reflector mode. Data were acquired and processed using Bruker flexAnalysis software where peaks were detected using the software's Snap algorithm. For relative quantitation of cyclic and linear peptides, the sum of the integrated peak areas of the first three isotopic peaks corresponding to each assigned peptide was taken as 100% of the expressed peptides. The percentage of cyclic and linear peptides within the sample could then be calculated.

**Peptide substrates**. The linear cyclotide precursor peptides kB1$_{GI}$ and kB1$_{GL}$ were chemically synthesized in-house by standard Fmoc solid-phase peptide synthesis. Folding and disulfide formation was carried out by incubating the reduced peptides in folding buffer (100 mM ammonium bicarbonate, 50% isopropanol, 2 mM reduced glutathione, 1 mM oxidized glutathione, pH 8.2) for 3 days [40]. The products were isolated by RP-HPLC at >95% purity and characterized by high-resolution MS and NMR spectroscopy. The model peptide R1$_{AEP}$ was supplied by GL Biochem at >85% purity, as determined by RP-HPLC and MS.

**Statistical analysis**. One-way ANOVA followed by Tukey's multiple comparisons test was performed using GraphPad Prism version 7.0c for Mac OS X, GraphPad Software, La Jolla California USA, www.graphpad.com. The predictive power of the covarying residue property set for binary classification of the known ligases and proteases was assessed using the metrics Specificity (true negative prediction rate by model), and Accuracy (proportion of correct predictions by model). These were calculated as Specificity = $TN/(TP + FN)$ and Accuracy = $(TP + TN)/(TP + FP + TN + FN)$, where TP is true positive, FP is false positive, TN is true negative, and FN is false negative.

**Data availability**. RNA-seq reads generated from *H. enneaspermus* and *P. x hybrida* "Mitchell" are available in the Sequence Read Archive with the following accession codes: SRR6394743 and SRR6394745 for *H. enneaspermus*; SRR6394744 for *P. x hybrida*. GenBank accessions for AEP gene sequences from *P. x hybrida* and *H. enneaspermus* are as follows: PxAEP1 (MG720071), PxAEP2 (MG720075), PxAEP3a (MG720072), PxAEP3b (MG720076), HeAEP-1 (MG720073), HeAEP-2 (MG720070), HeAEP-3 (MG720074). All other data supporting the findings of this study are available from the corresponding authors upon request.

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

## Acknowledgements

The pEAQ vectors were kindly provided by Prof. George Lomonossoff at the John Innes Centre and Plant Bioscience Ltd. We acknowledge funding from the Australian Research Council (ARC Laureate Fellowship (FL150100146) to D.J.C. and ARC grant DP150100443 to D.J.C., E.K.G., and T.D.). This research was also supported by the 2015 Ramaciotti Biomedical Research Award to D.J.C. and M.A.A. and by Hexima Ltd.

## Author contributions

M.A.J., E.K.G., T.S., K.S.H., T.D., D.J.C., and M.A.A. conceived the experiments. M.A.J., S.P., and H.J. made gene constructs, performed transient assays and MS analysis, E.K.G. prepared plant transcriptomic data and cloned AEP sequences. Q.K. performed the molecular modeling. T.S performed the AEP protein sequence space modeling. K.S.H., K.Y., and R.G. produced and assayed recombinant AEP. All authors contributed to the writing of the manuscript.

## Additional information

**Competing interests:** The authors declare no competing interests.

