## [Peer Review File · Nature Communications]

Reviewers' comments:

Reviewer #1 (Remarks to the Author):

Reviewer 1 was asked to evaluate the transcriptomic data, which were deemed adequate for the purpose of the manuscript.

Reviewer #2 (Remarks to the Author):

The authors report a comprehensive, well-balanced investigation of plant asparaginyl endopeptidases (AEPs) with peptide ligase activity, which are peculiar to a few groups of plants, and appear to have arisen in these plants by convergent evolution.

The ms is very well written and very straightforward: the authors compared diverse AEPs by sequence and sequence space analysis, compared activity of molecules in vivo/planta by means of transient expression of enzyme and target peptide in a non-ligase-expressing host, set about investigating by means of directed mutation / sequence swapping which were the residues / predicted structures responsible for the ligase activity, and used the information to identify - and test - hitherto unknown AEP ligase candidates predicted from sequence searches of databases.

I find their results highly convincing: they have established a good basis for predicting ligase activity in hitherto uncharacterised plant species; they used predictions to engineer activity in previously inactive enzymes; they established a very useful in planta system for testing candidates. Their results will almost certainly have immediate application in both the search for novel cyclised peptides, as well as in the deliberate engineering of plants to produce large quantities of pharmaceutically relevant cyclic peptides. I have no hesitation in recommending publication of this work.

Reviewer #3 (Remarks to the Author):

The manuscript by Jackson et al describes the structural features of AEPs that enable peptide ligation/cyclization versus proteolysis. The authors used a variety of techniques that include protein engineering, molecular modeling, and in silico protein space analysis to identify the residues key for the ligase activity. The authors proved this by introducing these mutations into proteolytic AEPs to improve their ligase efficiency.

In summary, the manuscript is well written and the results reported are quite interesting to the scientific community, therefore making suitable for publication after addressing the concerning comments.

Other comments:

- 1) Page 4, line 98. The sentence in the last paragraph seems too long (7 lines long!), which makes it difficult to follow.
- 2) Page 6, line 125. The authors claim that petunia AEP was unable to cyclize the native precursor of petunias cyclotide phybB in planta (using tobacco plant) due to the inability of the precursor to fold correctly. However, the kalata B1 precursor was obtained in planta under the same conditions in a 30% yield. Do the authors have any experimental evidence for this claim? After all it is well known that cyclotide kalata B1 is hard to fold in physiological conditions requiring the presence of organic solvents to fold correctly. Also, showing a figure comparing the sequences of these two cyclotides would be helpful to see the major sequence differences among these two cyclotides. In any case, to test this should be easy to check, for example using MS to test the presence of disulfides and/or qualitative NMR to prove or disprove a cyclotide-like structure.
- 3) Figure 2A. The sequences for polymorphisms of PxAEP3a and PxAEP3b shown in Figure 2A is a little bit confusing. The residue position for each should be probably added to the residue.
- 4) Since around 8% of the cyclization when tested in planta using the tobacco plant comes from endogenous AEPs, it would make sense to subtract as background activity when reporting it, for example in Figs. 2 and 3?
- 5) Based on the explanation provided in the text for the lack of cyclization activity for hAEP, due to glycosylation despite of having the MLA deletion, did the authors try to mutate that residue to Ala and test the activity of the corresponding mutant? Also, please specify how hAEP was obtained as is not clearly described in the experimental section.
- 6) The activity of butelase I under the conditions used in the manuscript is marginally or non-existent. Do the authors have an explanation?
- 7) The fact that all the ligase activities were tested in a kalata B1 substrate (and some in an SFTI-1 substrate) makes generalization a little more challenging. The authors should acknowledge that. It is specially puzzling the fact that petunia AEP ligases cannot cyclize petunia cyclotides in using the tobacco planta assay.

Reviewer #4 (Remarks to the Author):

Asparaginyl endopeptidases (AEPs) also known as legumains and vacuolar processing enzymes (VPEs), are endopeptidases that specifically recognize Asp or Asn residues. There are several papers showing the AEPs from some plant species have peptide ligation activity. It is seemed that there is no phylogenetic relationship among AEPs with peptide ligation activity. Therefore, solving how AEPs acquired peptide ligation activity will be important to understand mechanism of enzyme reaction of AEPs and the result will give an insight into the development of peptide ligation type of AEP. In the paper, the authors try to address this issue by experimental or computer analysis, and they found that amino acids near active center ('Gatekeeper' residues) and an adjacent loop determine the peptide ligase activity in plant AEPs. This finding is important because it makes easy to detect the peptide ligase type AEPs. It may explain how the two different types of AEPs occurred in distant plant species.

The authors' claim is interesting but I think careful interpretation will be required because recently two research papers from other groups show that peptide ligation activity of AEPs is simply depend on the environmental pH (Haywood et al, eLife vol. 7: e32955, Zauner et al, Plant Cell, in press, doi:10.1105/tpc.17.00963). I was impressed from those two papers that basically all AEPs have peptide ligation activity but the optimal conditions for the ligation activity are varied among AEPs. This is more acceptable for me than thinking multiple simultaneous convergent evolutions of ligase type AEPs from distant species. I think the authors' claim will not explain the basic mechanism for peptide ligation of AEP, but rather explain the optimal peptide ligation activity in vivo is accompanying with the changes of 'Gatekeeper' residues and lost of loop peptides in AEPs.

The two related papers in eLife and Plant Cell will be officially published prior to this paper but the claim in this paper is different from those papers. Therefore, it will be worth to publish but may confuse the research field. The authors will be required to describe more precisely to explain their findings. Specific comments are follows,

1. The main problem of the manuscript is the authors did not check the expression level of native or modified AEPs in heterologous expression system in plants (Figure 1b, 2d 3b, 5e, 6). There is

suspicion that the differences of cyclic peptide accumulation are the consequence of the difference of enzyme accumulation levels rather than enzyme specificity in heterologous system. I think proteinase activity based labeling (e.g. Biotin-xVAD-fmk) will work to estimate AEP levels in plants.

2. The authors are considering quite small numbers of AEPs are peptide ligase type with their own rules (lines 244-248). However, recent findings suggest that HaAEP1 (Haywood et al, eLife vol. 7: e32955) and AtLEGγ (Zauner et al, Plant Cell, in press, doi:10.1105/tpc.17.00963) have peptide ligation activity in more neutral pH. HaAEP1 and AtLEGγ have MLA, a protease type AEP signature, therefore incorporation of these findings in the authors' *in silico* analysis will compromise the estimation of ligase type AEPs. Furthermore, the authors do not include jack bean CeAEP1 (Bernath-Levin et al, Chem. Biol. vol 22: 571-582) as ligase type AEP in the analysis. The authors should explain the suitability for the selection of ligase type AEPs more convincible. I agree that authors can successfully pick up some of ligase type AEPs from AEP homologues, but I do not think that authors' method is enough to determine ligase or peptidase type AEPs. I think it is need to mention in the manuscript that the authors' method is a conservative way to find ligation type AEPs but the method may not applicable to some AEP homologues, especially peptidase type AEPs.

3. I think classical phylogenetic tree will be easier to understand than Supplemental Fig 11. There is also no information for nodes.

Reviewer 1 was asked to evaluate the transcriptomic data, which were deemed adequate.....

RESPONSE: Thank you, we appreciate the assessment of reviewer 1.

Reviewer #2

The authors report a comprehensive, well-balanced investigation of plant asparaginyl endopeptidases (AEPs) with peptide ligase activity, which are peculiar to a few groups of plants, and appear to have arisen in these plants by convergent evolution...I find their results highly convincing: they have established a good basis for predicting ligase activity in hitherto uncharacterised plant species; they used predictions to engineer activity in previously inactive enzymes; they established a very useful in planta system for testing candidates. Their results will almost certainly have immediate application in both the search for novel cyclised peptides, as well as in the deliberate engineering of plants to produce large quantities of pharmaceutically relevant cyclic peptides. I have no hesitation in recommending publication of this work.

RESPONSE: Thank you, we appreciate the assessment of reviewer 2.

Reviewer #3

The manuscript by Jackson et al describes the structural features of AEPs that enable peptide ligation/cyclization versus proteolysis. The authors used a variety of techniques that include protein engineering, molecular modeling, and in silico protein space analysis to identify the residues key for the ligase activity. The authors proved this by introducing these mutations into proteolytic AEPs to improve their ligase efficiency. In summary, the manuscript is well written and the results reported are quite interesting to the scientific community, therefore making suitable for publication after addressing the concerning comments. Other comments:

1) Page 4, line 98. The sentence in the last paragraph seems too long (7 lines long!)....

RESPONSE: Thank you. We have reworded the sentence for clarity.

2) Page 6, line 125. The authors claim that petunia AEP was unable to cyclize the native precursor of petunias cyclotide phybB in planta (using tobacco plant) due to the inability of the precursor to fold correctly. However, the kalata B1 precursor was obtained in planta under the same conditions in a 30% yield. Do the authors have any experimental evidence for this claim? After all is well known that cyclotide kalata B1 is hard to fold in physiological conditions requiring the presence organic solvents to fold correctly. Also, showing a figure comparing the sequences of these two cyclotides would be helpful to see the major sequence differences among these two cyclotides. In any case, to test this should be easy to check, for example using MS to test the presence of disulfides and/or qualitative NMR to prove or disprove a cyclotide-like structure.

RESPONSE: We were also surprised to observe this. In the original text, we suggested that this **might** be due to misfolding. Experimental evidence for this, however is difficult to obtain, as plants carry a sophisticated ER quality control process that rapidly removes any misfolded proteins. An assessment of the protein folding machinery differences between petunia and *N. benthamiana* would be informative, but we believe beyond the scope of the key message of this manuscript. Nevertheless, we have re-worded the sentence (p6 line 125) to make it clear that no PhybA related peptides (misfolded or not) were detected. As suggested by the reviewer, we have added the PhybA peptide sequence into Fig 1 aligned with that of kalata B1 and have explained the differences in the revised caption.

3) Figure 2A. The sequences for polymorphisms of PxAEP3a and PXAEP3b shown in Figure 2A is a little bit confusing. The residue position for each should be probably added to the residue.

RESPONSE: We have updated Figure 2 to include a residue alignment number for each polymorphism and updated Supplementary Figure 3 so that the readers can easily identify the residues in question.

4) Since around 8% of the cyclization when tested in planta using the tobacco plant comes from endogenous AEPs, it would make sense to subtract as background activity when reporting it, for example in Figs. 2 and 3?

RESPONSE: We respectfully disagree as the introduction of an exogenous AEP may in fact outcompete any endogenous AEP activity. We believe it is best to report the values obtained for each treatment.

5) Based on the explanation provided in the text for the lack of cyclization activity for hAEP, due to glycosylation despite of having the MLA deletion, did the authors try to mutate that residue to Ala and test the activity of the corresponding mutant? Also, please specify how hAEP was obtained as is not clearly described in the experimental section.

RESPONSE: Mammalian AEP (legumain) exists as a single isoform where a predominant role in proteolysis is reported (Chen, JM. et al. J Biol Chem. 272(12):8090-8). We did not endeavor to express the human legumain in plant cells, nor did we produce recombinant enzyme for functional analysis as we were focusing on plant AEPs sourced from cyclotide producing plant species. However, that said, without the other accessory residue changes (eg. polyproline loop, Gatekeeper residues) we don't think that just changing the MLA to a non-glycosylated form would by itself improve ligase capability of the human legumain.

6) The activity of butelase-I under the conditions used in the manuscript is marginally or non-existent. Do the authors have an explanation?

RESPONSE: The manuscript was submitted with *in planta* data showing that co-expression of butelase-1 improved the production of cyclic peptide (see Figure 3 b), and thus, we assume the reviewer meant butelase-2? This enzyme has been extensively characterized using recombinant produced enzyme and shows no ability to cyclise peptide substrates (Serra, A. et al. Sci. Rep. 6, (2016)). This gene was included in this analysis to serve as a control to compare against butelase-1. To make this clear we have added "the hydrolysis preferring" butelase-2 (p9 line 204) and added the appropriate reference for the recombinant characterization of butelase-2.

7) The fact that all the ligase activities were tested in a kalata B1 substrate (and some in an SFTI-1 substrate) makes generalization a little more challenging. The authors should acknowledge that. It is specially puzzling the fact that petunia AEP ligases cannot cyclize petunia cyclotides in using the tobacco planta assay.

RESPONSE: The reviewer is correct in that our *in planta* assessment of cyclisation is limited to cyclisation of kB1 and SFTI-1. In our recombinant enzyme work, however (Fig 5c and d), we additionally tested a non-cysteine containing model peptide substrate R1 with the results mirroring that of kB1. We also do mention in the discussion the possibility that the ligase type MLA may indeed have evolved to suit selection of cyclotide substrates. We also mention that despite this butelase-1 has been shown to cyclise a variety of peptide and protein substrates. Please see response above to petunia cyclotides.

Reviewer #4

Asparaginyl endopeptidases (AEPs) also known as legumains and vacuolar processing enzymes (VPEs), are endopeptidases that specifically recognize Asp or Asn residues. There are several papers showing the AEPs from some plant species have peptide ligation activity. It is seemed that there is no phylogenetic relationship among AEPs with peptide ligation activity. Therefore, solving how AEPs acquired peptide ligation activity will be important to understand mechanism of enzyme reaction of AEPs and the result will give an insight into the

development of peptide ligation type of AEP. In the paper, the authors try to address this issue by experimental or computer analysis, and they found that amino acids near active center ('Gatekeeper' residues) and an adjacent loop determine the peptide ligase activity in plant AEPs. This finding is important because it makes easy to detect the peptide ligase type AEPs. It may explain how the two different types of AEPs occurred in distant plant species. The authors' claim is interesting but I think careful interpretation will be required because recently two research papers from other groups show that peptide ligation activity of AEPs is simply depend on the environmental pH (Haywood et al, eLife vol. 7: e32955, Zauner et al, Plant Cell, in press, doi:10.1105/tpc.17.00963). I was impressed from those two papers that basically all AEPs have peptide ligation activity but the optimal conditions for the ligation activity are varied among AEPs. This is more acceptable for me than thinking multiple simultaneous convergent evolutions of ligase type AEPs from distant species. I think the authors' claim will not explain the basic mechanism for peptide ligation of AEP, but rather explain the optimal peptide ligation activity in vivo is accompanying with the changes of 'Gatekeeper' residues and lost of loop peptides in AEPs. The two related papers in eLife and Plant Cell will be officially published prior to this paper but the claim in this paper is different from those papers. Therefore, it will be worth to publish but may confuse the research field. The authors will be required to describe more precisely to explain their findings.

RESPONSE: The peptide ligase field is currently a hot topic. We have added a new paragraph in the discussion (p. 18, line 408-423) to describe the two recent studies and contrast our findings which we believe are unique.

Specific comments:

1. The main problem of the manuscript is the authors did not check the expression level of native or modified AEPs in heterologous expression system in plants (Figure 1b, 2d 3b, 5e, 6). There is suspicion that the differences of cyclic peptide accumulation are the consequence of the difference of enzyme accumulation levels rather than enzyme specificity in heterologous system. I think proteinase activity based labeling (e.g. Biotin-xVAD-fmk) will work to estimate AEP levels in plants.

RESPONSE: We have considered the possibility of using an activity based probe to estimate AEP levels in plants. However, this would give a measure of the sum total of AEP activity in the sample and not the contribution of an individual AEP (this heterogeneity is further complicated by glycosylation as well as differential autoprocessing (autoactivation) of AEPs). It is also unclear if the substrate mentioned (which was designed with Asp at P1 – for caspases) would be recognized equally between AEPs from different species or those that we know have different activity preferences. We have addressed this issue by monitoring transcript levels via RT-qPCR and have provided a further Supplementary Figure (now Sup Fig. 9) that demonstrates only minor differences in transcript abundance between OaAEP1, OaAEP2, OaAEP2+ and OaAEP2select. Furthermore, we prepared several AEPs recombinantly in *E. coli* (OaAEP1b, OaAEP2, OaAEP1b_MLA and OaAEP2+) and tested their activity in *in vitro* cyclization assays (see Figure 5c, d). The *in vitro* activity preference (ligase/protease) for each of these enzymes closely mirrored the *in planta* results (compare Figure 5d and e).

2. The authors are considering quite small numbers of AEPs are peptide ligase type with their own rules (lines 244-248). However, recent findings suggest that HaAEP1 (Haywood et al, eLife vol. 7: e32955) and AtLEGy (Zauner et al, Plant Cell, in press, doi:10.1105/tpc.17.00963) have peptide ligation activity in more neutral pH. HaAEP1 and AtLEGy have MLA, a protease type AEP signature, therefore incorporation of these findings in the authors' *in silico* analysis will compromise the estimation of ligase type AEPs. Furthermore, the authors do not include jack bean CeAEP1 (Bernath-Levin et al, Chem. Biol. vol 22: 571-582) as ligase type AEP in the analysis. The authors should explain the suitability for the selection of ligase type AEPs more convincible. I agree that authors can successfully pick up some of ligase type AEPs from AEP homologues, but I do not think that authors' method is enough to determine ligase or peptidase type AEPs. I think it is need to mention in the manuscript that the authors' method is a conservative way to find ligation type AEPs but the method may not applicable to some AEP homologues, especially peptidase type AEPs.

RESPONSE: The reviewer is right that we set our own strict criteria to call an AEP either a ligase-type or a protease-type based on our in-house kB1 *in planta* cyclisation assay. We established these criteria to be sure that the protein sequence space analysis was not contaminated with false positives, which would compromise the data, as suggested by the reviewer. The sunflower HaAEP1 is indeed (even at its more ligase favorable pH of 6.5) a poor ligase (see fig 4c of Haywood et al, eLife vol. 7: e32955; for the SFTI-1_NGLDN substrate roughly 10-20% cyclic product) and little was known of AtLEGy at the time the sequence space analysis was performed. AtLEGy comes from Arabidopsis, which is known not to be able to produce cyclic peptides (Gillon D et al., 2008, The Plant Journal, 53; 505-515), and therefore it was not considered as a potential ligase. We did not include jack bean CeAEP1 because, similarly to HaAEP1, it displays poor ligase activity *in vitro* (see Fig 6 Bernath-Levin et al, Chem. Biol. vol 22: 571-582) and it was also not tested with our preferred test substrate (kB1). We appreciate the stringent *in vivo* definition of our enzyme characterization and the reviewer's concern that we are overstating the importance of the MLA. In light of this, we have updated the discussion to include the possibility that cyclotide-producing plants may also contain AEPs with protease-type MLAs that may have a pH dependent functional role (see page 18)

3. I think classical phylogenetic tree will be easier to understand than Supplemental Fig 11. There is also no information for nodes.

RESPONSE: We agree that the network diagram, which was included to show the relationships between different types of AEPs, could be less familiar to most readers than a phylogenetic tree. We have swapped the network diagram in Supplementary Figure 12 (previously 11) for a phylogenetic tree that includes human (outgroup; HsAEP1), jackbean (CeAEP1), and cloned AEPs that are discussed in this manuscript. The resulting tree has a reduced complexity but still clearly shows that AEPs cluster into two broad groups: the Asterids (showed using red text) and Rosids (blue text). Ligase-type and protease-type AEPs are intermixed in the Asterids, which include the Rubiaceae and Solanaceae plants. The AEPs from Asterids cluster by plant family rather than by type of activity, suggesting that the distinction between the two activities evolved independently within the two families. By contrast, butelase-1 and (CtAEP1) and HeAEP3, are the two ligase-type activity AEPs within the Rosids, and these two AEPs are closely related according to this phylogeny. Node information (as percent of trees supporting each node) are given in the updated Supp Fig 12. The tree is rooted with human AEP (HsAEP1) and is a consensus neighbor-joining tree of 1000 bootstrapped trees.

REVIEWERS' COMMENTS:

Reviewer #4 (Remarks to the Author):

The main problem in the previous version of manuscript was that there is no verification in the expression level of native and modified AEPs in plant expression systems. In the revised manuscript, the authors added information about expression by RT-PCR. Although it is not the direct determination of protein levels, which I expected, but I agree that activity based estimation of AEP levels in plants will be difficult because the cross reaction of endogenous AEPs and the substrate specificity of enzymes as mentioned by the authors. They confirmed expression level were at least not so much difference among the AEP variants in plants. The authors adequately addressed the other comments in the revised manuscript.